# From the Kinetic Theory of Gases to the Kinetics of Rate Processes: On the Verge of the Thermodynamic and Kinetic Limits

**DOI:** 10.3390/molecules25092098

**Published:** 2020-04-30

**Authors:** Valter H. Carvalho-Silva, Nayara D. Coutinho, Vincenzo Aquilanti

**Affiliations:** 1Divisão de Modelagem de Transformações Físicas e Químicas, Grupo de Química Teórica e Estrutural de Anápolis, Centro de Pesquisa e Pós-Graduação. Universidade Estadual de Goiás, Anápolis 75132-903, Brazil; 2Instituto de Química, Universidade de Brasília, Caixa Postal 4478, Brasília 70904-970, Brazil; 3Dipartimento di Chimica, Biologia e Biotecnologie, Università di Perugia, 06123 Perugia, Italy; 4Istituto di Struttura della Materia, Consiglio Nazionale delle Ricerche, 00133 Rome, Italy

**Keywords:** Maxwell–Boltzmann path, Euler’s formula for the exponential, activation, transitivity, transport phenomena

## Abstract

A variety of current experiments and molecular dynamics computations are expanding our understanding of rate processes occurring in extreme environments, especially at low temperatures, where deviations from linearity of Arrhenius plots are revealed. The thermodynamic behavior of molecular systems is determined at a specific temperature within conditions on large volume and number of particles at a given density (the thermodynamic limit): on the other side, kinetic features are intuitively perceived as defined in a range between the extreme temperatures, which limit the existence of each specific phase. In this paper, extending the statistical mechanics approach due to Fowler and collaborators, ensembles and partition functions are defined to evaluate initial state averages and activation energies involved in the kinetics of rate processes. A key step is delayed access to the thermodynamic limit when conditions on a large volume and number of particles are not fulfilled: the involved mathematical analysis requires consideration of the role of the succession for the exponential function due to Euler, precursor to the Poisson and Boltzmann classical distributions, recently discussed. Arguments are presented to demonstrate that a universal feature emerges: Convex Arrhenius plots (*super*-Arrhenius behavior) as temperature decreases are amply documented in progressively wider contexts, such as viscosity and glass transitions, biological processes, enzymatic catalysis, plasma catalysis, geochemical fluidity, and chemical reactions involving collective phenomena. The treatment expands the classical Tolman’s theorem formulated quantally by Fowler and Guggenheim: the activation energy of processes is related to the averages of microscopic energies. We previously introduced the concept of “transitivity”, a function that compactly accounts for the development of heuristic formulas and suggests the search for universal behavior. The velocity distribution function far from the thermodynamic limit is illustrated; the fraction of molecules with energy in excess of a certain threshold for the description of the kinetics of low-temperature transitions and of non-equilibrium reaction rates is derived. Uniform extension beyond the classical case to include quantum tunneling (leading to the concavity of plots, *sub*-Arrhenius behavior) and to Fermi and Bose statistics has been considered elsewhere. A companion paper presents a computational code permitting applications to a variety of phenomena and provides further examples.

## 1. Introduction

A basic task of current molecular science is to elucidate how the kinetic behavior of a physicochemical system manifests within the temperature range of its “life span”: thermodynamics has its focus on states of the system and the transition between them, while the study of the rate of evolution of processes is the objective of kinetics. In thermodynamics (as in mechanics), it is ubiquitous to face the balance among various types of energy being exchanged; the connection from the molecular to macroscopic energy levels requires averages over the myriads of ways of change of microscopic configurations that determine the progress of events. The situation in chemical kinetics is intrinsically not so sharp, not only because systems in movement are much harder to be studied than those in steady-state equilibrium. Currently, the techniques in experimental and theoretical kinetics have been advancing enormously (although at a much slower pace when compared with those of thermodynamics), due to progress on production and detection of molecular beams and on classical and quantum simulations of molecular dynamics.

Aspects related to the foundations of the kinetics of rate processes were elaborated recently in previous papers [1,2,3,4]. In [1], fundamental concepts concerning statistical distributions and reaction rate theory were presented, including the definition of transitivity, a function of absolute temperature denoted as γ(T), based on extensive phenomenology that is being accumulated; a subsequent paper [2] considered the historical background of developments of chemical kinetics, leading to the basic foundations through analysis of key mathematical ingredients; in [3], the formulations based on the concept of transitivity were compacted and applied to the description of several phenomena on the temperature dependence of rate processes beyond Arrhenius and Eyring; and finally in paper [4], companion of this one in this topical issue, a computational code is described and provided to calculate kinetics and related parameters in chemical transformations and transport phenomena.

The need emerges of differentiating developments from those employed in thermodynamics, in spite of the fact that the kinetic theory of gases by Maxwell (and later by Boltzmann) was formulated more or less contemporary to the thermodynamics of Carnot (and later of Clausius): their thermodynamic vision was later merged turning the Maxwell theory essentially in terms of Boltzmann–Gibbs distributions. Additionally, as a well-known matter of fact in the literature [5,6,7], the Arrhenius equation, basic to chemical kinetics, was suggested as an empirical adaptation of the thermodynamics of chemical equilibrium developed by van’t Hoff (ca. 1880).

In the present work, account will be given to how a derivation of a theory of rate processes from non-equilibrium distributions involves essentially steps that are usual in thermodynamics, specifically as far as averaging procedures are concerned. A specific feature here is that we are progressing in the spirit of the well-known Darwin–Fowler formulation [8,9,10], which involves departure from the concept of “most probable” configuration emerging following the Boltzmann–Gibbs path [11,12]. Darwin and Fowler dealt with average quantities: they essentially developed a thermodynamics equivalent to the canonical form with no need of the concept of a microcanonical ensemble or even of that of entropy: a similar alternative path was briefly indicated by Eyring and coworkers presenting the foundations of the “Transition-State Theory” of rate processes [13,14,15,16]. This approach appears better motivated than the traditional: current experiments involve molecular beams studies of individual events, and advances in quantum mechanical treatments indicate the “royal path”: theoretical chemical kinetics proceeds by generating intermolecular potential energy surfaces and simulate computationally the passage from myriads of microscopic events to macroscopic quantities. This can be formulated at least in principle: however, when we consider polyatomic systems in molecular dynamics, we are unable to fully circumventing the difficulty presented by the need of averaging on a large number of events, difficult to be sampled in a statistically converged way. The situation was first anticipated by Maxwell who conjectured that the collective macroscopic observable motion of atoms, if they existed, should be compacted by averaging over statistical ensembles of their “trajectories” [17,18]. 

Following downward the upper part of the chart in Figure 1, we consider ab initio ‘exact’ quantum dynamics: it is expected to provide benchmark kinetics data, but is in practice still limited to simple cases, see [19,20,21]. Such applications of exemplary chemical reaction kinetics typically proceed according to the descending steps schematized in Figure 1: (a) calculation of the molecular electronic structure interactions involving high-level of quantum chemical accuracy, (b) dynamical evolution in phase-space configurations from the solution of the (usually time-independent) quantum equations of the motion, and (c) extraction of reactive properties from asymptotic scattering theory and calculating in succession key quantities: the quantum scattering matrix, the cumulative reaction probability and the cross sections. Finally, the Boltzmann weight averaging over a large span and fine grid of kinetic energies is needed to obtain the canonical quantity of chemical kinetics, the rate constants *k*(T) as a function of temperature. These rigorous prescriptions can yield benchmark results for quantum evolution of systems over a given potential energy surface and provide reaction rate constants for only a limited number of reactions: in fact, the complexity of the programming and the computationally demanding requirements strongly limit this type of study to reactive processes involving only a few atoms [19,21,22]. However, they serve as prototypes to complex molecular systems and stepping stones for example to processes governed by multiple potential energy surfaces, nonadiabatically coupled.

In this paper, the phenomenological rate theory [1,2,3] is developed by introducing a mechanism where the delay or acceleration of the approach to a well-defined mathematical limit due to Euler accounts for the low-temperature deviations of rates from Arrhenius law. In the next section, revisitation of the classical thermodynamic limit accompanies its extension to kinetics and naturally leads to a deformation of the Boltzmann–Gibbs distribution and to the emergence of a formulation alternative to that of Arrhenius [1,2,3]. Section 3 illustrates how theory serves to the natural scaling of a variety of physical and chemical processes in extreme conditions and near phase transitions. Implementations to various phenomena are sketched in Section 4. Concluding remarks are given in Section 5. Appendix A presents formulas for the distribution of energies in a reactive process away from equilibrium.

## 2. Thermodynamic versus Kinetic Limits, Revisited

### 2.1. The Exponential as Limit of Euler’s Succession: Role in the Early Kinetic Theory of Gases

The memorable succession for the exponential function from the sum of an infinite series is a powerful variant of the binomial theorem of Newton and was discovered by Euler in the XVIII century. For its occurrence originated in a famous bank account problem solved by Jacob Bernoulli and for aspects of its remarkable facets, see the recent papers [1,2,3,23]. The tremendous advances in the kinetic theory of gases started in the mid-XIX century with Maxwell’s mathematically intuition to look at the microscopic world as composed of greatly many indivisible particles, atoms. According to this vision, which found in Ludwig Boltzmann [24] one of the greatest defenders in times when even the existence of atoms was being questioned, the germs of what is now known as statistical mechanics were formulated: the motion of microscopic particles was correlated to macroscopic observables providing the foundations for the phenomenology of thermodynamics. It is not always recognized that the statistical proposition for success or failure of events (Bernoulli urn or Bernoulli trial binomial process and its generalization) provided through the Euler’s limit the foundations for the derivation of extensions to distributions, i.e., the foundations for the progress in the XIX and the early XX century, remarkably those of Poisson, Gauss, Planck, Bose–Einstein, and Fermi–Dirac [1].

At the very origin of the statistical mechanics viewpoint, the investigations reported in the 1860 [17] and 1866 [18] papers by James Clerk Maxwell lead to the famous velocity distribution of molecules under the hypothesis of the independence of Cartesian components of the velocity vectors: this conjecture appeared plausible from the additive properties of the exponential function. In 1868, Ludwig Boltzmann [25], as reviewed, e.g., in reference [26], introduced probabilistic concepts—the “marginal” probability of the energy of a molecule—obtaining a derivation of the Maxwell’s law of velocities by rigorous treatment based explicitly on the exponential behavior of velocity according to the Euler’s succession, see Figure 2.

A decade after, Maxwell [27] returns to the Boltzmann’s formulation proposing a more insightful approach of the problem, see Figure 3, rarely considered in the expositions of the theory in wide number of papers, treatises, and textbooks. In 1916, Jeans in his treatise on “The Dynamical Theory of Gases” [28] addresses Maxwell’s latest treatment within a much more concise mathematical analysis generalizing the concept of phase-space: again the exponential velocity distribution law is obtained from the Euler’s limit of a succession. This same procedure can be traced in further [29] and recent [30] works, where cases involving finite systems are dealt with essentially by arresting the treatment before taking Euler`s limit, namely without taking what is now recognized as the “thermodynamic limit”.

In the 7^th^ Chapter of his 1938 treatise “Kinetic Theory of Gases”, Kennard [31] develops a comprehensive assessment of macroscopic irregular motion of molecules (including, e.g., the Brownian motion) as connected to averaged microscopic fluctuations: the connection between discrete statistical distributions and exponential functions is obtained by the Euler’s succession, taking the limit to infinity of the number of particles. Earlier, in a collection of his investigations on statistical mechanics collected in a 1927 book, Tolman [32] had briefly discussed the role of taking limits in the description of fluctuations for a large number of molecules; in his treatise in 1938 [33] the theme of fluctuations and thermodynamic equilibrium are discussed in more details through a detour involving the Stirling formula for factorials and maximization of entropy in the Boltzmann–Gibbs approach. In either way these treatments involved imposing limiting values to specific variables and anticipating the operation that we will discuss in the next section, namely that of taking the thermodynamic limit, see [31,34]: in some cases, as intermediate steps in the course of derivations, physically insightful expressions were encountered.

### 2.2. The Thermodynamic Limit: The Contribution of Fowler and Collaborators

To give a general foundation to statistical mechanics, stepping stones can be schematized as follows. Darwin and Fowler [10,35] developed their approach in the early twenties, introducing the concept of temperature as the zero principle and defining as key quantities specific distributions and in particular partition functions [10,36]. In a lucid lecture, in 1948 Schrödinger [37] describes their achievements as major after those of Boltzmann (1868) [25] and Gibbs (1902) [12]. A few years later a mathematical analysis around the concept of the thermodynamic limit was carried out by Yang and Lee [38,39,40]: they considered the limit as to be taken with respect to properties in the neighborhood of phase transitions, and gave a deep theory of associated analytic singularities.

Basic to a variety of modern treatments, the thermodynamic limit concept was mentioned as central in many ways: there are at least three recent books [41,42,43] and a dedicated paper on statistical mechanics [44] where the treatments are from the very beginning based on the introduction of the concept of thermodynamic limit. Reference [44] refers to the Darwin–Fowler method as powerful alternative to the Boltzmann–Gibbs celebrated construction and describes the Bogoliubov contribution. Technically, Darwin and Fowler [8,9] and Fowler and Guggenheim [35] obtain average quantities from multivariable distributions using an asymptotic method, that is of the steepest descent: reference [45] on p. 53 shows equivalence with taking the Stirling limit for factorials and the Lagrange maximization of functions by undetermined multipliers, a procedure which is standard in the Boltzmann–Gibbs statistical approach to entropy.

Usually, in most popular books the thermodynamic limit is defined only in words. The quantitative definition [46,47] is provided considering extensive variables, the volume *V* and the number of particles *N* going to infinity while their ratio, the density *ρ* = *N*/*V*, remains finite: see Figure 2 for the reproduction of the original treatment by Boltzmann. The formulas exploit essentially the limit of a succession due to Euler to obtain the exponential function [2]; in books by both Pathria [42] and Huang [40] on statistical mechanics, the very first concept introduced is the thermodynamic limit and provide accessible qualitative, useful presentations of the contributions by Yang and Lee [38,39,40].

In one of the Landau and Lifshitz series of textbooks, there is a treatment now considered as standard [46]. The section arguably written by L.P. Pitaevski, addresses the problem of fluctuations, obtaining the Poisson distribution considering the volume *V* of gas occupied by a number of particles *N*. Let *v* be a small part of the total volume and proceeding with the same ingredients used by Boltzmann in 1868 ([25] and Figure 2) one can show that the probability for a volume *v* to contain *n* molecules follows a Jacob Bernoulli’s binomial distribution of the type
(1)$$P\left(n\right)=\frac{1}{n!}\frac{N!}{\left(N-n\right)!}{\left(\frac{𝓋\;}{V}\right)}^{n}{\left(1-\frac{𝓋\;}{V}\right)}^{N-n}.$$

Taking the limit
(2)N→∞ and  V→∞
for an average number of particles $\bar{n}$ in volume V, while the density, namely the ratio
(3)$$\rho =\frac{N}{V}=\frac{\;\bar{n}}{𝓋}$$
remains finite, the passage from the binomial distribution to the Poisson distribution is obtained by the Euler’s formula for the exponential function as a limit of a succession (see also the references [1,41]):(4)$$P\left(n\right)=\frac{{\bar{n}}^{n}}{n!}e^{-\bar{n}}$$

This derivation can be taken as representative of the content of the expression “taking the thermodynamic limit”.

Other treatments are worthy of mention. Using a path analogous to Boltzmann’s “marginal” probability, Eyring and collaborators [14,15,16] provide an elementary presentation based on a paper by Condon (1938) [34] on a statistical mechanics derivation of the Boltzmann distribution law. The treatment is interesting for chemical kinetics. Considering the equilibrium of a molecular subsystem within a system composed of *s* harmonic oscillators [15], it is possible to calculate the probability of the molecular subsystem to acquire an energy ε. Assuming a total energy *E* of the system, one can identify the number of ways to distribute $m=\frac{E-\varepsilon }{h\nu }$ quanta of energy among the *s* oscillators of the system (*hν* is the energy per quantum):(5)$$W\left(s,m\right)=\left(\begin{array}{c} s+m-1\\ m \end{array}\right)=\frac{\left(s+m-1\right)!}{m!\left(s-1\right)!}.$$

The probability of the molecular system to acquire the energy ε is
(6)$$P\left(s,m\right)=\frac{\left(\begin{array}{c} s+m-1\\ m \end{array}\right)}{\sum_{j}\left(\begin{array}{c} s+j-1\\ j \end{array}\right)}.$$

Taking what is now recognized as the thermodynamic limit (in this case, s and m tending to infinity) and using the Euler’s formula for the exponential function as a limit of a succession, the Boltzmann law is recovered
(7)$$P\left(\varepsilon_{i}\right)=\frac{e^{-\frac{\varepsilon_{i}}{k_{B}T}}}{\sum_{j}e^{-\frac{\varepsilon_{i}}{k_{B}T}}}.$$
where *ε* is energy, *k_B_* is Boltzmann’s constant, and *T* the absolute temperature. In reference [1] modifications needed to obtain Bose–Einstein and Fermi–Dirac distributions are sketched.

As we have seen, it went unnoticed that many formulations had been anticipated by Boltzmann in his 1868 article [25]. Indeed, he himself in the famous paper published in 1877 [1] changed focus, and developed the celebrated procedure: that of searching for most probable values with limits on particle numbers at a given total energy to obtain the entropy by the Lagrange method of undetermined multipliers. Due to this spectacular result the attention of most subsequent investigations was diverted away from the kinetic approach towards thermodynamic treatments. However, in 1940 Hinshelwood [48] sketched a pedagogical justification of Boltzmann’s exponential distribution considering the probability of favorable collisions that can lead to a specific energy accumulation into molecules: he relies explicitly on the Euler’s limit to obtain the exponential function. See more details in [2], where it is emphasized that when interpreting this mechanism as that operating in the activation of molecules one has a profound insight on chemical reactivity and on rate processes.

### 2.3. Avoiding the Thermodynamic Limit Describes Nonlinearities of Arrhenius Plots

In chemical kinetics the difficulty occurs in finding macroscopical and canonical kinetics properties, ingredients analogous to those of thermodynamics. Temperature can be introduced assuming the zero principle for the preparation of reactants: the question arises how to use an analogue of the thermodynamic limit when it is arbitrary to define extensive quantities like number of particles (*N*) or volume (*V*) in a reactive process.

For kinetics of the chemical and physical rate processes the evidence [49,50,51,52,53,54,55,56] of deviations from Arrhenius behavior is increasing—arguably it is to be associated to moving away significantly from the Boltzmann–Gibbs distribution, particularly going down to low temperatures. In Boltzmann’s equation of Figure 2 substituting *k*/*x* with *ε*^‡^*β*/*N* we performed the same mathematical operation as for the thermodynamic limit: the passage of the distribution when *n* goes to infinity can be written:(8)$$P_{N}\left(\varepsilon^{‡}\beta \right)={\left(1-\frac{\varepsilon^{‡}\beta }{N}\right)}^{N}\overset{N\to \infty }{\to }P\left(\varepsilon^{‡}\beta \right)=e^{-\varepsilon^{‡}\beta }$$
where *ε*^‡^ (the activation energy) represents an energetic obstacle for the process to occur and *β* = 1/*k_B_T* is the usual Lagrange multiplier, the “coldness” [2,57]. The exponential Boltzmann–Gibbs distribution *P*(*ε*^‡^*β*) emerges as a limiting case of a power law distribution *P_N_*(*ε*^‡^*β*) we have shown [1,58,59,60] that the latter, corresponding to avoiding taking the limit permits that the low temperature deviations in kinetic processes can be described with remarkable consistency in a generality of contexts. This treatment makes explicit the connection of *P_N_*(*ε*^‡^*β*) distribution with Tsallis statistics [61,62] identifying 1/*N* with 1 − *q*, where *N* is allowed to be continuous and Tsallis *q* is classically limited in a small range.

The final expression in (8) is Arrhenius law apart from the pre-exponential factor *A*: Further convenient introduction of the deformation parameter *d* in place of 1/*N* one has
(9)$$k_{𝒹}\left(\varepsilon^{‡}\beta \right)=A\;{\left(1-𝒹\varepsilon^{‡}\beta \right)}^{1/𝒹}\;\overset{d\to 0}{\to }\;k\left(\varepsilon^{‡}\beta \right)=e^{-\varepsilon^{‡}\beta }$$

The left-hand side of the correspondence (9) is known as the Aquilanti–Mundim deformed Arrhenius formula. We amply proved that it could be considered uniformly both for *d* < 0 (quantum propensity) and for *d* > 0 (classical propensity). The first case has been treated amply elsewhere [59,63,64]; we mostly focused here on the second case [60,65]. Rarer cases are found for which *d* > 0 and $\varepsilon^{‡}$ < 0, and are referred as corresponding to an *anti*-Arrhenius behavior [66,67,68].

There are some examples in the literature [29,69,70,71] where there is evidence that our language the classical thermodynamic limit is reached due to the magnitude of Avogadro number $O\left(10^{24}\right)$. Considering this number, the statistical mechanics treatment indicates that fluctuations of energy in a canonical ensemble turn out extremely sharp and narrow. This peculiarity of the energy distribution is required to admit the equivalence between averages and most probable values of variables: applicability of statistical techniques to the foundations of thermodynamics relies on this kind of argument, upon which is also based the concept of the thermodynamic limit. At low temperatures, especially in reactive and non-reactive processes, conditions can be violated since the order of magnitude of the activated events is controlled by the Eyring pre-exponential factor 2πℏβ~ O(1013) with a consequent possibility of relaxation of the thermodynamic limit.

Far away from the thermodynamic limit at extreme conditions, a lower thermal limit can be assumed for this *super*-Arrhenius case where chemical or physical rate processes would require an infinite time to proceed, i.e., when
(10)$$1-𝒹\varepsilon^{‡}\beta \approx 0$$
this condition provides an interpretation for the parameter *d* already pointed out and defined by us in previous papers [1,2,3] (and recently confirmed [72,73]):(11)$$𝒹=\frac{k_{B}T^{†}}{\varepsilon^{‡}}=\frac{\varepsilon^{†}}{\varepsilon^{‡}}$$
where the superscript † denotes a minimum temperature *T*^†^ or a thermal energy threshold *ε*^†^ for which the kinetic process is operative. The *d* parameter differs from zero as a scale to measure the “thermal limit of propensity” with reference to the lower and higher kinetic energy values respectively to *ε*^†^ and *ε*^‡^, see Scheme 1: (i) when β→0, and also remaining finite $𝒹=\frac{\varepsilon^{†}}{\varepsilon^{‡}}$ (Cauchy’s limit [73]) Equation (9) also tends to the exponential law; and ii) when β→∞, $𝒹\varepsilon^{‡}\beta \to 1$ and $𝒹=\frac{\varepsilon^{†}}{\varepsilon^{‡}}$ remaining finite, a minimum limit can be identified for which the kinetic process may occur. Case ii) applies for *d* > 0, the *super*-Arrhenius cases: the *sub* limit is consistent with Wigner’s threshold law for thermoneutral reactions [74,75,76]. Furthermore, taking advantage of an alternative expansion [77,78] for the Aquilanti–Mundim law
(12)k(β)=(1−𝒹ε‡β)1𝒹=Ae−ε‡β[1−12𝒹(ε‡β)2−13𝒹2(ε‡β)3−18(2𝒹−1)𝒹2(ε‡β)4+O(β5)]

Note a misprint in Equation (17) in reference [3]. The three limits previously described are illustrated as defining the thermodynamic and kinetic limits, summarized in the following scheme (see also Figure 4).

From Figure 4, it is possible to perceive graphically the three limits in a 3D Arrhenius plot, ln*k vs*. *d* and *β*. When *d* tends to zero the Arrhenius law is recovered. For *d* > 0, a convex curvature is generated (*super*-Arrhenius) and the tendency for a lower thermal limit is observed. When *d* < 0 the plot becomes concave (*sub*-Arrhenius) because of quantum mechanically tunneling.

For small tunneling, we showed that [63]
(13)𝒹=−13(hν‡2ε‡)2
where *ε*^‡^ is the barrier height, directly proportional to ν^‡^, the square of the frequency for crossing the barrier at the maximum in the potential energy surface. For the concave case, the tendency is attenuated and known as the Wigner limit [74,75] for thermoneutral chemical reactions (see in Scheme 1):(14)lim∀𝒹; 𝒹ε‡β→1A (1−𝒹ε‡β)1𝒹=A β1𝒹.

### 2.4. Architecting the Transitivity Concept

From a conceptual viewpoint and with reference to Figure 1, in a previous paper [2] we emphasized how essentially a statistical mechanics path to chemical kinetics (the theory of change) can be based on a theory of chance, where however criteria for choices need be provided [2,3]: the transitivity concept is exhibited as playing a crucial role.

A recent paper [3] also gives an account of how useful it is the introduction of representations of experimentally or numerically exact rate constant data; the first, of course, is the Arrhenius plane [5], whereby the apparent activation energy is interpreted according to the Tolman’s theorem [79]; the second one is the transitivity plane. We sketched here and elaborated elsewhere [4] a further aspect justifying how the definition of the transitivity function that can provide an understanding of microscopic kinetic processes using alternative forms of scaling of the rate data, yielding naturally the conventional statistics used in rate process, from Maxwell–Boltzmann [34] to Tsallis statistics [62], and to the further popular Vogel–Fulcher–Tammann [80,81,82] distributions.

#### 2.4.1. Tolman’s Theorem and the Apparent Activation Energy

Conventionally, starting points in the statistical thermodynamics proposed by Willard Gibbs [12] and Fowler et al. [10,35] are the average energy *E* of a canonical system obtained as the logarithmic derivative of the partition function *Z* with respect to *β*
(15)$$\bar{E}=-\frac{d}{d\;\beta }\ln Z$$

When we turn to kinetics, the correspondence can be established considering the average energy to be accumulated by colliding molecules to proceed to reaction. Following Tolman (the first paper is one hundred years old [79]), well-characterized is the concept of apparent activation energy *E_α_*(*β*). This entity is customarily obtained by chemical kinetics data on reaction rate coefficients *k*(*β*) k(β), phenomenologically from the Arrhenius plot (as recommended in 1996 by the definition of the International Union of Pure and Applied Chemistry [83]):(16)$$E_{a}\left(\beta \right)=-\frac{d}{d\;\beta }\ln k\left(\beta \right)$$

The apparent activation energy can be written as the difference between the average internal energy of the reacting molecules and that of all molecules in the system: this is the content of statement of the so-called Tolman’s theorem [35,79], which has been analyzed quantum mechanically by Fowler and Guggenheim [35]: the meaning is that *E_a_* represents a measure of an energetic obstacle to the progress of the reaction, reinterpreted subsequently and exploited as the barrier height energy in Eyring’s formulation of the transition state theory [13]. Basic is to consider the apparent activation energy as essentially the *ε*^‡^ parameter of the previous section (actually, the double dagger notation is that introduced by Eyring).

#### 2.4.2. Planck Black-Body Radiation and Reciprocal Energy

We provided now a further perspective viewpoint of the phenomenological parameters involved in the definition of the activation energy and its reciprocal, the transitivity function, going back to the elementary formulation of Planck to solve the problem of the average energy of a black body [14,84]. Assuming a system composed of harmonic oscillators with quantized energy $\epsilon_{n}=nh\nu$, where *ν* is the frequency of the oscillator, the partition function for this system is given as,
(17)$$Z=\sum_{n}e^{-\epsilon_{n}\beta }$$

The total average energy of this system can be calculated using Equation (15)
(18)$$\bar{E}=-\frac{d}{d\;\beta }\ln\underset{n}{\sum }e^{-\epsilon_{n}\beta }\;\;\bar{E}=-\frac{d}{d\;\beta }\ln\left[1+e^{-h\nu \beta }+{\left(e^{-h\nu \beta }\right)}^{2}+{\left(e^{-h\nu \beta }\right)}^{3}+\ldots \right]\;\;\bar{E}=-\frac{d}{d\;\beta }\ln\left[\frac{1}{1-e^{-h\nu \beta }}\right]=\frac{h\nu }{e^{h\nu \beta }-1}$$

At low temperatures (β→∞), expansion of the reciprocal of $\bar{E}$ assumes a functional form that produces a power law in *β*,
(19)$$\frac{1}{\bar{E}}=\frac{1}{h\nu }\left[1+h\nu \beta +\frac{1}{2!}{\left(h\nu \beta \right)}^{2}+\frac{1}{3!}{\left(h\nu \beta \right)}^{3}+\frac{1}{4!}{\left(h\nu \beta \right)}^{4}+\ldots \right]$$
suggesting the usefulness of introducing also to this case the inverse of $\bar{E}$, as an analog of the transitivity function *γ*(*β*). See [1] for elaboration of quantum statistical treatments.

#### 2.4.3. Activation and Transitivity: A Prototypical Unimolecular Reaction Model

Another interesting case that generates a functional form justifying the introduction of transitivity is the model considered in the Twelfth Chapter of reference [35] by Fowler and Guggenheim: they propose the initial steps of a kinetic theory for unimolecular processes, starting from a quantum theoretical formulation of Tolman’s theorem for calculating the probability for molecules to react after acquiring a sufficient amount of energy *ε*^‡^ distributed over the *s* internal degrees of freedom characterizing the reacting molecule. The model, arguably valid within a small enough neighborhood of *β* = 0, is that all active molecules have the same probability of decomposition, requiring an average over all possible values of energy for the active molecules: these hypotheses represent the prototypical theory of unimolecular reactions further elaborate into that of Slater [85] and to the more successful RRKM formulation. We find interesting to elaborate further and provide a continuation of their formulation.

Consider, as first suggested by René Marcelin [86], that a molecule is “active” if the energy exceeding *ε*^‡^ is distributed over s internal vibrational degree of freedom: the last formula of Fowler and Guggenheim presentation for the unimolecular rate constant can be transcribed here in our notation in a remarkably simplified form
(20)$$k\left(\beta \right)=Ae^{-\varepsilon^{‡}\beta }\overset{s-1}{\underset{r=0}{\sum }}\frac{1}{r!}{\left(\varepsilon^{‡}\beta \right)}^{r}.$$

First, we note that the sum can be given in closed form
(21)$$k\left(\beta \right)=A\left(\frac{e^{\varepsilon^{‡}\beta }\Gamma \left(s,\varepsilon^{‡}\beta \right)}{\Gamma \left(s\right)}\right)e^{-\varepsilon^{‡}\beta }.$$
where $\Gamma \left(s,\varepsilon^{‡}\beta \right)$ is the incomplete Γ-function. Additionally, we calculated the activation energy for the Fowler–Guggenheim model by logarithmic differentiation of the rate coefficient (21) with respect to β
(22)$$E_{a}=-\frac{\partial \ln k}{\partial \beta }=\varepsilon^{‡}-\frac{\sum_{r=1}^{s-1}\frac{1}{\left(r-1\right)!\beta }{\left(\varepsilon^{‡}\beta \right)}^{r}}{\sum_{r=0}^{s-1}\frac{1}{r!}{\left(\varepsilon^{‡}\beta \right)}^{r}}=\varepsilon^{‡}\frac{{\left(\varepsilon^{‡}\beta \right)}^{s-1}}{\Gamma \left(s,\varepsilon^{‡}\beta \right)}e^{-\varepsilon^{‡}\beta }.$$

Finally, we got a closed form in β also for the transitivity function
(23)$$\gamma \left(\beta \right)=\frac{1}{E_{a}}=\left(s-1\right)!\overset{s-1}{\underset{r=0}{\sum }}\frac{1}{r!\beta }{\left(\varepsilon^{‡}\beta \right)}^{r-s}=\frac{1}{\varepsilon^{‡}}\frac{\Gamma \left(s,\varepsilon^{‡}\beta \right)}{{\left(\varepsilon^{‡}\beta \right)}^{s-1}}e^{\varepsilon^{‡}\beta }.$$

This model for unimolecular reactions is of interest for any pseudo-first-order processes. Two other cases analyzed by reference [35] should be similarly investigated, permitting asymptotic analysis through well-known properties of special functions.

## 3. Scaling in the Transitivity Plane

### 3.1. Transitivity and Renormalization Group Coupling

It is important to recognize the similarity between the functional form of transitivity *γ* with respect to the rate coefficient *k* and the reciprocal temperature *β* (Figure 1); and the renormalization group coupling parameter, credited to Callan [87] and Symanzik [88] (see also Wilson [89]): $\beta_{CS}\left(g\right)=-(\partial g/\partial \ln\mu )$ defines modernly the relationship between the coupling constant *g* and the energy scaling function *μ*. The equation encodes the mathematical apparatus in both quantum field theory and the theories of critical phenomena used to handle problems with singularities, such as those occurring at phase transitions. See the lucid presentation by Weinberg [90] (see also [91,92]).

### 3.2. Classes of Universal Behaviors

The previous sections have shown that from a kinetic point view, the reciprocal of the activation energy can be properly defined as the transitivity *γ*, specific of a process and interpreted as a measure of the propensity for the reaction to proceed. Our notations stem from the fact that the transitivity can take a gamma of values smooth as function of *β* in a sufficiently ample range of temperatures. Its limiting values will serve to localize any abrupt changes, e.g., in mechanisms of processes or in phase transitions. Generally, if a Laurent expansion defined in references [1,2,3,4] is assumed to hold in a neighborhood around a reference value denoted as *β*_0_, it behaves asymptotically as
(24)$$\gamma \left(\beta \right)\;\sim \;\alpha \;{\left(\beta -\beta_{0}\right)}^{\zeta }$$

General series for *γ*(*β*) where previously given in reference [1,3]. Now, the transitivity plane, *γ vs*. *β*, (see Figure 3) can be interpreted as confining the range of existence of a system between limiting temperatures in consonance with the thermal kinetic limits defined in Section 2.2. The two temperatures or limiting coldnesses β are generally contained between the extremes *β* ≈ 0 to *β* ≈ *β*^†^ defining the temperature window where a process is operative. The simplest model for γ is a linear path from *α* = 1/*ε*^‡^ to *β*^†^= 1/*ε*^‡^ according to the AM formula [3]. In fact, the limiting formula derived from Equation (24) in reference [3] yields
(25)$$\gamma \left(\beta \right)=\frac{1}{\varepsilon^{‡}}\left(1-\frac{\beta }{\beta^{†}}\right).$$

It is interesting to express known temperature-dependence rate laws generalizing the previous equation as
(26)$$\gamma \left(\beta \right)=\frac{1}{\varepsilon^{‡}}{\left(1-\frac{\beta }{\beta^{†}}\right)}^{\zeta }$$

The exponent ζ = 0, 1, and 2 generates the Arrhenius, Aquilanti–Mundim (AM), and Vogel–Fulcher–Tammann (VFT) laws, respectively [3]. Many other paths can serve as models for the transitivity function for different values of ζ (see Figure 5). Generalization to non-integer values shows perspectives of correlation with critical exponents in mode coupling theory and with universality classes of kinetic transitions (see also Section 4). Studies in the glass transition field show [93,94,95] that systems with a large fragility (strong non-Arrhenius behavior) present ranges of universality separated by a crossover temperature: in some works considering glass-forming systems and, e.g., for the prototypical reaction F + H_2_ (D_2_) at low and ultra-low temperatures [20], should permit to categorize the universality classes in a wide temperature range by the critical exponent ζ, possibly empirically a non-integer.

In reference [3], we show cases when curvature in the Arrhenius plot can be linearized. Interestingly, a formulation was empirically proposed in 1980s to fit the temperature dependence of properties of glass-forming materials [97,98,99]. Here, the proposed connection through the Tolman’s theorem is assumed as a scaling tool for relaxation processes: the relationships appearing in the transitivity plane turn out as explicitly universal for the linear dependence in *β*, at least in a significantly wide neighborhood near the origin of β≿0. Most important is that all parameters are given both a physical meaning and the possibility of being estimated by physical models.

## 4. Perspectives on Rate Processes from the Arrhenius and the Transitivity Planes

There are a variety of chemical reactions and rate processes that deviate from Arrhenius behavior, and this list of them is currently expanding upon consideration of several types of phenomena being documented [1,59,60,100,101]. Below, we collected three important examples in which a systematic investigation of the universality is in progress.
(i)The rates of biological processes are strongly affected at low temperatures by deviations from Arrhenius law; however there are large uncertainties especially when quantifying, as usual these deviations using the “Arrhenius Break Temperature” assumption, see previous discussions [3,60]. The difficulty of identifying a transition temperature in the Arrhenius plot for the respiration processes [60,102,103] can be easily overcome using the transitivity plot, emphasizing sudden transitions described within the Aquilanti–Mundim law (universality class with ζ = 1).(ii)Further applications concern the glass transition phenomenon occurring in a variety of materials: This is considered one of the most complex open problems in condensed matter physics. In the neighborhood of the glass transition temperature, the kinetic coefficients—diffusion, viscosity, and relaxation time—present deviations from the Arrhenius law specifically depending on the material composition. In reference [4], we examined the nonlinear temperature dependence of the relaxation time of propylene carbonate [98,99] from the transitivity plot: it is presented a perspective tool to observe a transition temperature connecting regimes described by two Aquilanti–Mundim straight lines in transitivity plane, and identifying the crossover temperature [93,94].(iii)Among phenomena akin to glass transitions but on extremely larger timescales, very important are those occurring in geochemical environments, where nonlinearity of the temperature dependence of the viscosity of rocks is often observed in the Arrhenius plots. In Figure 6, the nonlinearity in Arrhenius plot for **Cl_OF** silicate [49,104,105] also obeys the Aquilanti–Mundim law when analyzed in the transitivity plot: however, no transition temperature is revealed in this case.


In the exemplary processes presented above, the scaling provided by the transitivity variable makes explicit the corresponding extent of deviations from Arrhenius law, emphasizing kinetic transition temperatures when they appear. Computational tools for assisting in assessing this behavior are presented in a code described in a companion article in this topical collection [4].

## 5. Conclusions and Outlook

The approaches to the description of rate processes formulation have evolved due to the synergism between phenomenological and computational approaches: with Arrhenius law representing the former and the transition-state theory standing at the foundations of the latter. However, with the advance of experimental and computational techniques, these approaches needed extensions able to cope with new problems, such as quantum effects (e.g., tunneling and resonance) in atomic and molecular systems, stochastic motion of particles in condensed environment, non-equilibrium effects in classical and quantum formulations. From several modern techniques for treating kinetic problems, we can cite Feynman-like path integral formulations [106,107,108] to estimate temperature dependence of rate constants in chemical reactions, mode-coupling theory [109] for describing the physics of glass formation; and the development of rational extended thermodynamics [23,110] to treat systems far away from equilibrium.

The implementation of modern formulations to new experimental data and computational simulations requires a complex set of microscopic information to estimate kinetic parameters, making formidable the problem of describing many-particle dynamics and kinetic equations. Concomitantly, phenomenological approaches continue to be important pillars for the enhancement of ab initio formulations beyond: Arrhenius [5], Vogel–Fulcher–Tammann [80–8582 Williams–Landel–Ferry [111], Power law [112], Bässler [113], and Nakamura–Takayanagi–Sato [114] laws are a sequence of useful models to describe problems in extreme and highly complex environments. Motivated by this perception and establishing connection with Tsallis statistics [62] for classical propensities, in the last ten years we have worked in close synergism between phenomenological and ab initio or semiempirical formulations. A key guide came by Euler’s expression for the exponential function as a limit of succession, a formulation accompanied by physicochemical meanings originally suggested for gas kinetic theory and chemical kinetics processes.

The “prequel” to the saga has been reconstructed in Section 2. We recapitulate the steps that originated essentially from following the Maxwell–Boltzmann path and involving at some stage application of the Euler’s formula: Boltzmann (1868) [25] (Figure 2) was the first that succeeded to prove the Maxwell’s distribution working with marginal probabilities in what is now called the thermodynamic limit; subsequently, Maxwell (1879) [27] (Figure 3) and Jeans (1916) [28] developed rigorous formulations performing mathematically the Euler’s formula for the thermodynamic limit; Uhlenbeck and Goudsmit [29] in their study of finiteness of particle number stated clearly that their formulation is written in the spirit of the Maxwell–Boltzmann original treatments. In these formulations at the final stage always the Euler’s limit is invoked, by which the exponential distribution function is recovered upon taking the thermodynamic limit.

Connection with concomitant modern approaches is relevant. Recently, it has been asserted that the molecular world and its reactivity can be interpreted by theories involving Fuzzy sets and Fuzzy logic [115]. These theories have been formulated by the electronic engineer Lotfi Zadeh, and are useful to model how humans “compute” by using words [116]. Every word of the natural language, represented by a Fuzzy set, is like a “quantum” of information, whose meaning is context-dependent. Similarly, every molecule or every atom of the microscopic world is like a Fuzzy set, i.e., like a word of the “molecular language”. Every molecule can exist as a collection of different conformers and every atom as a superposition of different quantum states. These “molecular Fuzzy sets” show context-dependent behaviors.

The connection with modern statistical mechanics appears to emerge as follows. In our context, it is needed to understand the meaning of the deviations from the Arrhenius equation. Let us cite from the incipit of [41]:

In statistical mechanics we are concerned with the physical properties of large systems. We assume the existence of the thermodynamic limit (a main concern in this paper, Section 2). The peculiarity, which requires that the mechanics of such a system is “statistical”, stems from the fact that such a system is as a rule incompletely defined. By this we mean that the equations of motion for such a system cannot be uniquely solved. Were this true in Gibbs’ time already for the simple reason of mathematical complexity, the real problem is not computational, as is clear from interesting computer simulations currently available. Basically, the need for statistical methods stems from the lack of detailed information on the system.

The chain of emerging connections continues. The interpretation of the experimental evidence might require Zener`s geometric programming optimizations [117]: geometric programming is a nonlinear mathematical optimization method used to minimize functions that are in the form of polynomials subject to constraints of the same type. The connection between geometric programming and the Darwin–Fowler method has been established since some time [117] (see also a modern approach [118]). Since the data used in the optimization procedure are always affected by errors and uncertainties, a strategy to handle them is provided by the theory of Fuzzy sets, as discussed very recently [119], for example in reference [4], in generally in most of our work we used the generalized simulated annealing (GSA) [120]. The application of Fuzzy optimization algorithms can avoid rigidity and stiffness and reduce information loss arising from the conventional optimization procedures of statistical mechanics.

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
