# Peer review of "From the Kinetic Theory of Gases to the Kinetics of Rate Processes: On the Verge of the Thermodynamic and Kinetic Limits"

_molecules, 2020, doi:10.3390/molecules25092098_

Round 1

Reviewer 1 Report

Nice written paper. The extensive bibliography is very valuable. Nevertheless some problems, the paper is interesting, worth publication and broad discussion.

  1. Definition of beta=1/RT in Fig.1 shouls be repleaced by the same as in text (line 254)
  2.  "Table 2" in caption of Fig.2 is not clear at all.The same problem is with "figure 51 and continuous on [Eq.55]" in caption of Fig.3. In fact both Figures (2 and 3) are extremally interesting.
  3. Conection of d>0 or d<0 with quantum propensity and clasical propensity is questionable (misliding). Some more broad discussion will be valuable (if such conclusion is correct)
  4. Interpretation for the parameter d (lines 280-281) needs more broad discussion (not olny reference).

The paper (with small revision)  can be recommended for publication.

Author Response

Comments and Suggestions for Authors

Nice written paper. The extensive bibliography is very valuable. Nevertheless some problems, the paper is interesting, worth publication and broad discussion.

Definition of beta=1/RT in Fig.1 should be replaced by the same as in text (line 254)

Done

 "Table 2" in caption of Fig.2 is not clear at all. The same problem is with "figure 51 and continuous on [Eq.55]" in caption of Fig.3. In fact both Figures (2 and 3) are extremally interesting.

Both captions were rewritten.

Connection of d>0 or d<0 with quantum propensity and classical propensity is questionable (misleading). Some more broad discussion will be valuable (if such conclusion is correct)Interpretation for the parameter d (lines 280-281) needs more broad discussion (not only reference).

Discussion expanded.

The paper (with small revision)  can be recommended for publication.

Reviewer 2 Report

In this article the authors not only review their own previous work on the extensions of Arrhenius equation for a large range of temperatures, but also they compile conclusions and remarks of historical and current articles from the begging of thermodynamics-kinetics-statistical mechanics in more than 120 references. Certainly, the article worth publishing. I foresee a considerable number of citations to this article from researchers in the field.

I would encourage the authors to give more examples instead of just referring to the topics where the theory is applicable.

Author Response

In this article the authors not only review their own previous work on the extensions of Arrhenius equation for a large range of temperatures, but also they compile conclusions and remarks of historical and current articles from the begging of thermodynamics-kinetics-statistical mechanics in more than 120 references. Certainly, the article worth publishing. I foresee a considerable number of citations to this article from researchers in the field. I would encourage the authors to give more examples instead of just referring to the topics where the theory is applicable.

The theoretical background is being formulated concomitant with the evaluation of experimental data. In section 4, we present examples where our formulation is being applied to different fields also beyond reaction kinetics: the companion paper and several references report the state of the art for the variety of applications of the theory and its implementation.
